# ON ACCURATE EVALUATION OF GANS FOR LANGUAGE GENERATION

## ABSTRACT

Generative Adversarial Networks (GANs) are a promising approach to language generation. The latest works introducing novel GAN models for language generation use n-gram based metrics for evaluation and only report single scores of the best run. In this paper, we argue that this often misrepresents the true picture and does not tell the full story, as GAN models can be extremely sensitive to the random initialization and small deviations from the best hyperparameter choice. In particular, we demonstrate that the previously used BLEU score is not sensitive to semantic deterioration of generated texts and propose alternative metrics that better capture the quality and diversity of the generated samples. We also conduct a set of experiments comparing a number of GAN models for text with a conventional Language Model (LM) and find that none of the considered models performs convincingly better than the LM.

## 1 INTRODUCTION

Neural text generation has achieved impressive results in the past few years Hassan et al. (2018); Wu et al. (2016). These models are typically conditional extensions of neural Language Models trained with the Negative Log Likelihood (NLL) objective to estimate probability distribution of the next word given a ground-truth history. While being very successful, they still suffer from a number of problems. Arguably, the most prominent are exposure bias Bengio et al. (2015) and a mismatch between the NLL objective used during training and a task-specific metric that we would like to minimize Bahdanau et al. (2016). Exposure bias stems from the fact that there is a mismatch between training and inference procedures. During training a model always receives histories that come from the well-behaved ground-truth sequences, whereas at inference it is conditioned on its own imperfect predictions.

Reinforcement Learning techniques that have recently received increased interest in the NLP community carry the promise of addressing both of these issues by allowing for task-specific (even non-differentiable) loss functions and incorporating sampling directly in the training process. However, previously used manually designed metrics based on n-gram matching such as BLEU Papineni et al. (2002) or ROUGE Lin (2004), are crude approximations for the true objective of generating samples that are perceptually indistinguishable from the real data.

The recently proposed Generative Adversarial Networks (GAN) framework Goodfellow et al. (2014) goes beyond optimizing a manually designed objective by leveraging a discriminator that learns to distinguish between real and generated data samples. It thus mitigates both issues of NLL training, since it includes sampling into the training procedure and aims at generating samples that cannot be discriminated from the real data points. Despite their recent success in the image generation domain (both unconditional Berthelot et al. (2017); Karras et al. (2017) and conditional Reed et al. (2016); Odena et al. (2017)), applying GANs to text generation is still a challenging task. One of the many challenges that slows down the progress is the lack of proper evaluation, which is a largely unsolved problem and is an active area of research. Previous works studying GANs for text generation have either reported metrics specific to a family of algorithms Semeniuta et al. (2017) or resorted to BLEU scores where a validation set is used as a reference Yu et al. (2016); Guo et al. (2017) to asses the quality of the generated samples. The main goal of this work is to study the currently adopted evaluation approach to GAN models for language generation, explore their shortcomings and propose novel solutions.

In particular, we demonstrate that previously used n-gram matching, such as BLEU scores, is an insufficient and potentially misleading metric for unsupervised language generation. Another issue that has so far been ignored by previous works on GANs for language generation is the sensitivity of models to the choice of hyperparameters and random seed initialization. We find that reporting results from the best single run or not performing sufficient tuning introduces significant bias in the reported results, which prevents researchers from making informed model choices. This becomes especially important in the presence of a quickly growing number of various GAN-related techniques.

Hence, in this work we focus on making a fair evaluation of neural generative models for text with a number of goals in mind. Firstly, we are looking for metrics that allow for a meaningful comparison. Secondly, we demonstrate the importance of reporting model sensitivity to multiple runs and hyperparameter choices in contrast to reporting a single best achieved score. Thirdly, we perform a set of experiments comparing multiple GAN models for text using our newly proposed protocol and metrics. We focus on GANs for multiple reasons. Firstly, as opposed to conventional Language Models and Variational Autoencoder (VAE) based models Bowman et al. (2015b); Semeniuta et al. (2017) that can be relatively easily compared with one another through perplexity values, comparing GANs with these models is difficult. Using n-gram statistics can be misleading and comparing based on perplexity puts GANs at a significant disadvantage since they do not optimize this objective. Secondly, GAN-based models can potentially solve both issues of NLL-based models discussed above.

**Our contributions are as follows:**

- We present an in-depth discussion of the problem of evaluation of unsupervised generative models of natural language.

- We demonstrate why previously used n-gram matching is an inadequate metric for language generation, propose alternatives, and validate their effectiveness.

- We propose a simple yet powerful comparison protocol for unsupervised text generation models that addresses training instability and gives a better picture of a model's behavior than comparing best achieved results.

- We study a number of GAN models for language generation using our proposed protocol and metrics and compare them with a conventional neural Language Model. Our main finding is that, when compared carefully, a conventional neural Language Model performs at least as well as any of the tested GAN models. When performing a hyperparameter search we consistently find that adversarial learning hurts performance, further indicating that the Language Model is still a hard-to-beat model.

## 2 RELATED WORK

Currently, the evaluation protocol adopted by the previous work on GAN-based text generation Yu et al. (2016); Guo et al. (2017) is primarily based on metrics derived from n-gram matching, e.g., BLEU and self-BLEU, which are used to assess sample quality and diversity. Additionally, a single best metric Yu et al. (2016); Guo et al. (2017) is reported which does not convey how sensitive various models are w.r.t. random initialization and hyperparameter choices. In this paper, we demonstrate that n-gram based metrics are inadequate for evaluation of unsupervised text generation models. Furthermore, we demonstrate that GAN models are extremely sensitive to random initialization and careful hyperparameter tuning is a must to have a meaningful comparison.

GANs are a promising algorithm specifically designed to generate samples indistinguishable from real data. This has led to an increased interest in systematic comparison of different algorithms, both for images Lucic et al. (2017) and texts Lu et al. (2018). A recent study Lucic et al. (2017) shows that careful hyperparameter tuning is very important for a fair comparison of different image GAN models and significant improvement can be achieved with a larger computational budget rather that from a better algorithm. Another recent work studies a set of GANs targeted specifically at text generation Lu et al. (2018). However, it has all the drawbacks of the accepted evaluation approach, namely using n-gram based metrics and reporting only a single best result. It is thus difficult to draw a convincing conclusion based on this kind of comparison. Another related work conducts a study on the properties of Variational and Adversarial Autoencoder-based generative models Cífka et al.

(2018). The main difference from this work is that we focus on GANs that are more difficult to compare than VAEs.

In this work we first address the issue of metrics used for evaluation of textual GANs. We then introduce a number of alternatives to the widely used BLEU and self-BLEU scores and demonstrate that they are capable of detecting a number of failure modes that the BLEU score does not capture. We then demonstrate the need of extensive hyperparameter tuning and conduct an initial set of experiments comparing a number of recent GAN-based approaches to text generation.

## 3 MODELS

### 3.1 CONTINUOUS MODELS

Continuous models for text closely follow how GANs are applied to images, i.e. they treat a sequence of tokens as a one-dimensional signal and directly backpropagate from the discriminator into the generator. We adopt the architecture of a continuous GAN model for language generation from Gulrajani et al. (2017). The generator is a stack of one-dimensional transposed convolutions and the discriminator is a stack of one-dimensional convolutional layers. The use of continuous generator outputs allows for straightforward application of GANs to text generation. To train this model we use the proposed WGAN-GP objective proposed by Gulrajani et al. (2017). The authors use a feedforward network as a generator, which consists of a stack of transposed convolutional layers (Conv-Deconv). Such a generator, however, does not properly model the sequential structure of language. Thus, we also consider an RNN-based generator. To ensure it remains continuous and gradients from D can be backpropagated into G, instead of taking argmax or sampling from the output distribution at each step, we feed the entire softmax output as the input for the next step of G. This follows the generation process of RNN-based Language Models with the difference that it models the conditional distribution $p(x_t|x_{<t})$ implicitly. Another option is to make use of annealed softmax temperature, gumbel softmax Jang et al. (2016) or straight-through estimator Bengio et al. (2013), but we leave it for the future research.

### 3.2 DISCRETE MODELS

Discrete models learn the distribution over the next token $p(x_t|x_{<t})$ explicitly and thus sample (or take argmax) from the output distribution at each step. This makes the generator output non-differentiable and gradients from D can no longer be backpropagated through G.

To train such a non-differentiable generator one can use Reinforcement Learning (RL) where scores from D are treated as rewards. The majority of discrete GAN models for text generation employ RL to train their models Yu et al. (2016); Guo et al. (2017); Fedus et al. (2018). However, in addition to instability of GAN training one has to also address problems of RL training such as reward sparsity, credit assignment, large action space, etc. For example, one approach to the credit assignment issue is to use Monte-Carlo(MC) rollouts Yu et al. (2016), which allows for providing a training signal to the generator at each step. Most commonly adopted solution to avoid reward sparsity is to pre-train the generator with the NLL objective, since sampling from a randomly initialized model in large action spaces makes it extremely hard to discover a well formed sequence. Many other RL techniques have been applied to NLP problems, for instance actor-critic methods, e.g., Bahdanau et al. (2016), or hierarchical learning Guo et al. (2017). However, these are out of the scope of this paper.

**SeqGAN.** In its simplest form RL-based GAN would employ a generator and a discriminator scoring the full sequence. The generator can then be trained with the REINFORCE objective $J_g = \sum_t D(y) * \log(p(y_t|y_{<t}))$. We refer to this variant as SeqGAN-reinforce. While this objective is enough in theory, in practice it has a number of problems. One such problem is credit assignment, where single per-sequence decision is an overly coarse feedback to train a generator. To address this, we consider two options. In SeqGAN-step we make use of the discriminator that outputs a decision at every step following previous research that has addressed credit assignment with this approach Fedus et al. (2018). The generator's loss is then given by $J_g = \sum_t R_t * \log(p(y_t|y_{<t}))$, where $R_t = \gamma * R_{t+1} + D(y_{1:t})$. Such a formulation allows us to more accurately perform credit assignment and make sure that the generator does not behave greedily and take into account the long term effect a generated token might have. The issue however is that scoring an incomplete sequence might be difficult. To address this we follow the SeqGAN model Yu et al. (2016) and employ

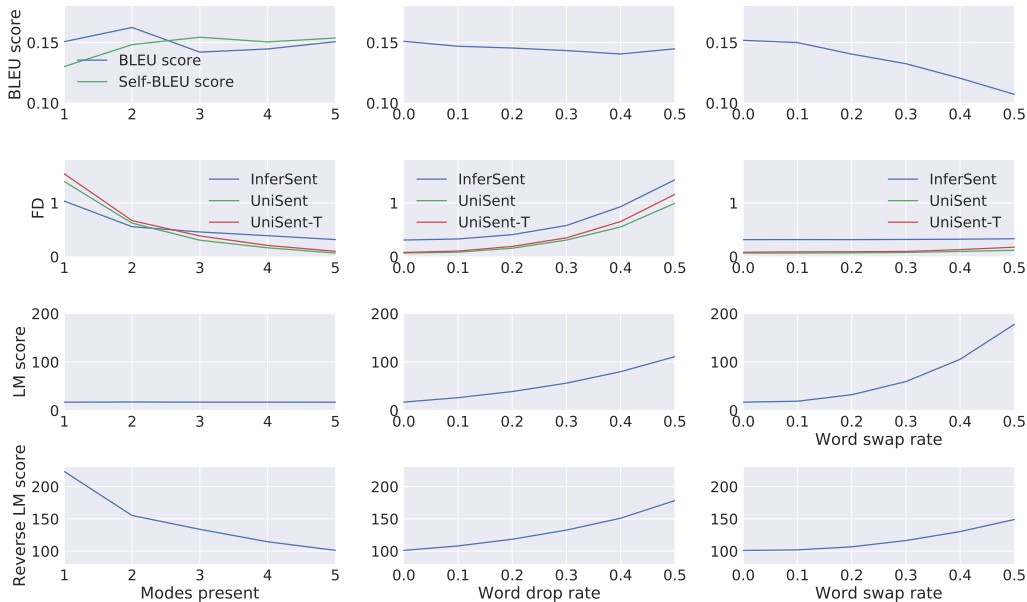

Figure 1: Scores assigned by four considered metrics for data with controllable amount of quality deterioration. Each row shows one metric and each column one task. Note that neither BLEU nor self-BLEU scores capture semantic deterioration of the data. For BLEU higher is better. For other metrics lower is better. We increase FDs obtained with UniSent and LM score embedding by a factor of 10 and decrease LM scores by the same factor for visualization purposes.

MC rollouts to continue a sequence till the end. We then score these rollouts with a per-sequence discriminator and use its output as a reward. We will refer to this variant as SeqGAN-rollout in the rest of the paper. The three considered variants are close to the original SeqGAN model and differ in their approach to the credit assignment problem.

**LeakGAN.** To address GAN instability in the RL training setup, a recent work Guo et al. (2017) proposes to reveal discriminator's state to the generator. We decouple this idea from the complicated RL training setup used by the authors and study the utility of passing discriminator's state to the generator during training. We consider three variants of the *LeakGAN* model that differ in how a hidden state of D is made available to G: *leak*, *noleak* and *mixed*, where the generator has access to the discriminator's, generator's and both hidden states respectively. Note that in LeakGAN-leak generator is an MLP and it does not maintain its own hidden state, only consuming that of the discriminator. Lastly, we do not update discriminator weights during generator's update phase. We note that these variants are simpler when compared to the original LeakGAN model Guo et al. (2017) since we do not use the RL technique used by the authors and do not interleave GAN and NLL training. These simplifications allow us to decouple the influence of the architectural changes from other dimensions. We refer the reader to Appendix B for further discussion of the model.

## 4 METHODOLOGY

### 4.1 METRICS

**N-gram based metrics.** Typical metrics that researchers have used to evaluate textual GANs are the number of unique n-grams Xu et al. (2018) and dataset level BLEU scores Yu et al. (2016); Guo et al. (2017). We use BLEU4 throughout the paper since we found the results to be similar for different sizes of N-grams. While they do give some insight into a model's behavior they have a number of drawbacks. The main criticism is that they do not capture semantic variations in generated texts and can only detect relatively simple problems with syntax. Other n-gram based metrics, e.g. ROUGE

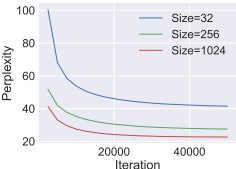 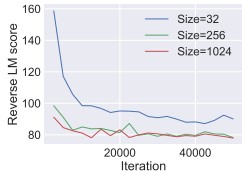 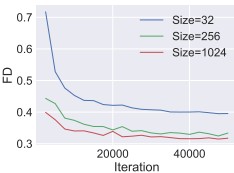

Figure 2: Learning curves of three differently sized Language Models. For all metrics lower is better.

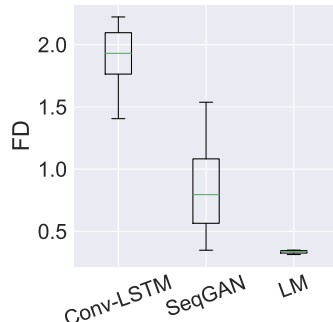

Figure 3: Distributions of FDs achieved by 30 best trials of three different models during hyperparameter search.

**Conv-LSTM GAN** (BLEU4=0.197, FD=1.464)

a young woman is sitting on a into into, on his sit
young woman woman woman while a group of son
the people are hair with sons
a little girl is wearing dogss
the children is at a red
man man white whiteing

**Language Model** (BLEU4=0.204, FD=0.273)

a man is competing in his ski class
the man is playing the accordion
she is the baby's sisters
the man is walking towards the fountain
a boy is climbing a tree lined
a man uses what looks to be a lawn mower

Table 1: Random samples from two models with close BLEU scores and considerably different FD.

and METEOR, would also suffer from similar issues. In this work we focus on BLEU since it is the accepted metric in the textual GANs community.

**Language Model score.** Another feasible way to evaluate a model is to estimate the likelihood of samples under a pretrained Language Model. This, however, has a drawback that a model that always generates a few highly likely sentences will score very well. Despite this, it is still a useful metric reacting only on the quality of generated samples and thus is a good proxy for a model's precision.

**Reverse Language Model score.** A more general approach is to train a Language Model on samples from a model and then evaluate its performance on a held out set of real texts Zhao et al. (2017). In this setting, however, the score is biased due to two factors. One is model bias caused by the imperfection of the LM that may not be good enough to model the data distribution. The other is data bias caused by the fact that we use a data sample to train a LM that will serve as a proxy for the true data distribution.

**Frechet InferSent Distance.** Another approach to evaluate a generative model is through an embedding model. Originally proposed in the computer vision community, Frechet Inception Distance (FID) Heusel et al. (2017) computes the distance between distributions of features extracted from real and generated samples. Inception refers to a specific image classifier architecture Szegedy et al. (2015). While researchers have pointed out that FID has its drawbacks, namely that it is a biased metric Lucic et al. (2017); Binkowski et al. (2018) and it makes unnecessary assumptions about feature distributions Binkowski et al. (2018), it is a widely accepted metric in the Computer Vision community. In this work we adapt this metric for text by using InferSent text embedding model Conneau et al. (2017), which is a bidirectional LSTM with max pooling trained in a supervised manner. Unless otherwise noted, we use InferSent embedding model to compute sentence embeddings. However, we note that training a sequence embedding model is an ongoing research which will likely affect the quality of the discussed metric. Thus we omit specific embedding model from the metric name refer to it as Frechet Distance (FD).

**Human Evaluation.** Since the goal of a text generation model is to create samples that are indistinguishable from real ones, it is important to also perform human evaluation to assess their quality. We

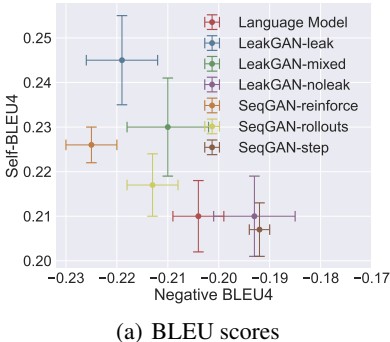
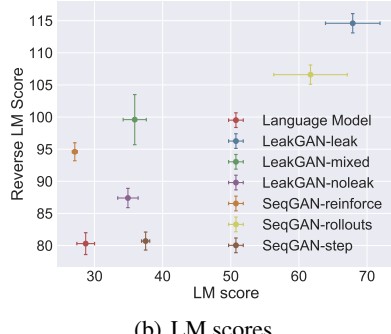

(a) BLEU scores          (b) LM scores

Figure 4: Results of best models shown on two complementary axes. We show negative values of BLEU4 for visualization purposes. Note that according to BLEU scores three models have comparable results, while LM scores show significantly better results for one model. We omit Conv-Deconv and Conv-LSTM models from these Figures since they show results considerably worse than those of other models.

send 200 samples from each model (uniformly sampling from random restarts) to the human raters (using 3 raters per sample) asking them to score if the presented sentence is grammatically correct and understandable on a scale from 1 to 5 (with being 5 the best score).

## 4.2 PARAMETER OPTIMIZATION PROCEDURE

Since GANs are very sensitive to the choice of hyperparameters, we optimize these parameters using random search limiting the computational budget to 100 trials. Once we have discovered the best performing set of hyperparameters, we retrain a model with these hyperparameters 7 times and report mean and standard deviation for each metric to quantify how sensitive the model is to random initialization. To justify the need of such a procedure we show distributions of results achieved by three models during one run in Figure 3. As expected, GAN-based models are considerably less stable than Language Model.

In addition, we generally see that the best results achieved in a run are usually obtained with a fortunate random seed supporting the second step of our procedure where we retrain a number of models and then report both mean and standard deviation using the best hyperparameters found during model tuning. We use the Adam optimizer Kingma & Ba (2014) to train our models and tune its hyperparameters separately for the generator and the discriminator. When training models with per-step discriminators we also tune the discount factor $\gamma$.

## 5 EXPERIMENTS

**Data.** We perform our experiments on the Stanford Natural Language Inference (SNLI) Bowman et al. (2015a) and MultiNLI datasets Williams et al. (2017). SNLI is a dataset of pairs of sentences where each pair is labeled with semantic relationship between two sentences. We discard these labels and use all unique sentences to train our model. The size of the resulting dataset is 600k unique sentences. We preprocess the data with the SentencePiece model with a vocabulary size equal to 4k. MultiNLI follows the SNLI structure but also provides a topic that a sentence pair comes from. This allows us to emulate mode collapse and thus measure the recall. We use SNLI for model comparison and MultiNLI for metric evaluation.

## 5.1 METRIC EVALUATION

**Mode collapse.** To emulate samples with varying degrees of diversity, we sample sentences from the train set using a fixed set of allowed topics. We then use the development set containing the full set of topics as a reference. An evaluation metric should be able to capture the fact that some topics,

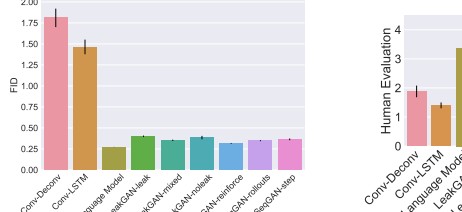 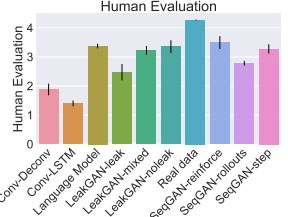 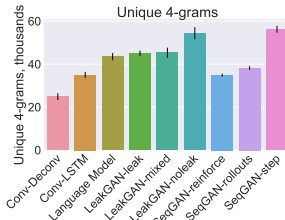

Figure 5: Results of models on FD, Human evaluation and Unique 4-grams.

e.g. fictional sentences, are present in the reference but not in samples. Results of this experiment are shown in Figure 1. We find that results vary when the number of topics is small, so we run the evaluation 5 times and report the average. Note that BLEU and LM score fail to capture semantic variations. FD, on the other hand, drastically increases as we remove more and more topics. This also holds for the reversed LM score. To test robustness of FD to the choice of embedding model we use two additional sequence encoders Cer et al. (2018) on the same data. One model is a mean pooled uni- and bigram embeddings followed by a feedforward network (UniSent). The other is a more computationally expensive Transformer Vaswani et al. (2017) based model (UniSent-T). The models are trained with a combination of supervised and unsupervised learning. We find that all three models show comparable results suggesting that FD is robust to the choice of a sequence embedding model. We also evaluate the self-BLEU score that has been used to evaluate the degree of mode collapse Lu et al. (2018). In this experiment, however, we observe that self-BLEU cannot detect this kind of mode collapse.

**Sample quality.** To measure metric sensitivity to the changes in the sample quality we introduce two types of perturbations in the samples. One is word dropout where we remove words with certain probability $p$ that controls the quality of the samples. The other is word swapping where we take a fraction of words present in a sentence and randomly swap their places. Results of these experiments are presented in columns 2 and 3 of Figure 1. Interestingly, the BLEU score is not very sensitive to word dropping. FD, on the other hand, significantly worsens under heavy word dropout. The situation is the opposite for word swapping, where the BLEU score is reacting more than FD. We attribute this behavior of FD to the underlying sequence embedding model. Since we use a bi-directional LSTM with max-pooling, it might have learned to be position-invariant due to pooling and is thus having difficulties detecting this kind of syntactic perturbations. Further research on better sequence embedding models is likely to improve the quality of evaluation with FD. LM score successfully captures decreased quality of samples but does not react to decreased diversity. Reversed LM score is sensitive to all three types of deteriorations.

In our second experiment we train three LSTM Language Models with one hidden layer with sizes 32, 256 and 1024. In this setting a larger model consistently achieves lower perplexity scores and thus we expect a metric to be able to detect that larger model produces better samples. In addition, we evaluate the models during training to also get the FD and LM score curves. Results of this experiment are shown in Figure 2. Note that all three metrics exhibit strong correlation and generally maintain ordering between differently sized models and different checkpoints of the same model. Our experiments suggest that both FD and reverse LM score can be successfully used as a metric for unsupervised sequence generation models. We generally observe reverse LM score to be more sensitive. However, it is prohibitively expensive to use during tuning. We thus opt for FD as a metric to optimize during hyperparameter searches.

## 5.2 GAN MODEL COMPARISON

For all GAN models that we compare we fix the generator architecture to be a one-layer Long Short-Term Memory (LSTM) network (except for the Conv-Deconv model). Other types of generators show promise Vaswani et al. (2017), but we leave them for further research.

Figures 4 and 5 show the results obtained by various models using our evaluation procedure. We make the following observations: (i) discrete GAN models outperform continuous ones, which could be attributed to the pretraining step – most discrete models barely achieve non-random results without supervised pretraining; (ii) SeqGAN-reinforce achieves lower LM score and higher human ratings than the Language Model but higher reverse LM scores, suggesting improved precision at large cost to recall; (iii) Most of GAN models achieve higher BLEU scores than the LM, while other metrics disagree, showing that looking only at BLEU scores would put the LM at a significant disadvantage; (iv) No GAN model is convincingly better than the LM. However, the LM is not convincingly better than SeqGAN-reinforce either. While the LM achieves lower FD, LM score and human evaluations prefer the GAN model. This further supports that it is important to report different metrics – reporting only FD would make the comparison biased towards the LM; (v) We do not observe improvements of models with access to the discriminator's state, suggesting that the previously reported good result Guo et al. (2017) may be due to the RL setup; (vi) Supervised pretraining of the generator is extremely important, since training of every GAN model that achieves reasonable results includes pretraining step. We refer the reader to Appendix A for a table presentation of these results.

To further demonstrate that BLEU scores are not representative of a model's quality we present samples from the Conv-LSTM GAN and the Language Model in Table 1. We make the following observations: Conv-LSTM GAN's samples are qualitatively worse than those of the Language Model due to spelling and syntactic errors. Its sentences are also generally less coherent. However, the difference in BLEU score between these two models is less than 1 point. It is thus difficult to draw conclusions from BLEU scores alone whether SeqGAN-rollout produces better samples than a Language Model since the difference in BLEU scores for these two models is also less than 1 point. FD and reverse LM score, on the other hand, reveal that samples from Conv-LSTM GAN are considerably worse than those from a Language Model.

Human evaluation supports FD and the reverse LM score and also assigns better scores to the Language Model. Note that in this particular case simply inspecting samples from Conv-LSTM GAN and a Language Model would suffice. We are, however, interested in automated comparison of models, where BLEU scores seem to not show reliable results.

# 6    CONCLUSIONS

In this work we focus on a proper evaluation of GANs for language generation. We have discussed drawbacks of previously adopted evaluation using BLEU scores and focused on the Frechet Distance and reverse Language Model scores. Our results suggest that BLEU scores are insufficient to evaluate textual GAN systems. In contrast, we have shown that both FD and reverse LM scores can successfully detect deteriorations that BLEU is not sensitive to. In addition, we have proposed a more systematic evaluation protocol and shown evidence that it provides a better picture than just reporting the single best result.

We used the proposed protocol and metrics to evaluate a number of adversarial text generation systems. We found that properly tuned conventional Language Models yield better results than any of the considered GAN-based systems. In fact, with proper hyperparameter tuning we find that when evaluated with FD the best results are achieved when the learning rate of the GAN generator after pre-training is the lowest, which corresponds to not performing GAN training at all, further supporting the need of reporting a number of metrics. These results generally agree with those obtained by a recent study Lu et al. (2018). The authors find that most models yield worse results than a simple Language Model. However, they do not perform hyperparameter tuning and report only BLEU scores, which makes it difficult to draw a convincing conclusion from the proposed comparison.

Our future work will be focused on a comparison of a larger array of GAN-based models for text generation. While we have performed an initial set of experiments comparing different GAN models, we do not claim it to be exhaustive. Indeed, when comparing a set of algorithms it is virtually impossible to take all possible sources of variation into account. It is thus possible that a well-behaved language GAN was not included in our search. Our future work will be focused on better disentangling various dimensions affecting the results of GAN models. We also aim at performing similar comparison for conditional models, such as image captioning and machine translation.

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

## APPENDIX A: RESULTS

| Metric | Language Model | Conv-LSTM | Conv-Deconv |
|---|---|---|---|
| Unique 4grams ↑ | $43.5k \pm 1.7k$ | $35k \pm 1.4k$ | $24.9k \pm 1.6k$ |
| BLEU4 ↑ | $0.204 \pm 0.005$ | $0.197 \pm 0.003$ | $0.08 \pm 0.02$ |
| Self-BLEU4 ↓ | $0.21 \pm 0.008$ | $0.34 \pm 0.02$ | $0.45 \pm 0.11$ |
| FD ↓ | $0.273 \pm 0.001$ | $1.464 \pm 0.087$ | $1.81 \pm 0.11$ |
| LM score ↓ | $28.7 \pm 1.3$ | $221 \pm 15$ | $2800 \pm 1100$ |
| Reverse LM score ↓ | $80.3 \pm 1.7$ | $2273 \pm 358$ | $4000 \pm 0.3$ |
| Human evaluation ↑ | $3.37 \pm 0.08$ | $1.4 \pm 0.1$ | $1.88 \pm 0.2$ |
| | SeqGAN-reinforce | SeqGAN-step | SeqGAN-rollouts |
| Unique 4grams ↑ | $34.9k \pm 0.7k$ | $56.2k \pm 1.6k$ | $38.2k \pm 0.8k$ |
| BLEU4 ↑ | $0.225 \pm 0.005$ | $0.192 \pm 0.002$ | $0.213 \pm 0.005$ |
| Self-BLEU4 ↓ | $0.226 \pm 0.004$ | $0.207 \pm 0.006$ | $0.217 \pm 0.007$ |
| FD ↓ | $0.316 \pm 0.005$ | $0.364 \pm 0.01$ | $0.348 \pm 0.006$ |
| LM score ↓ | $27.1 \pm 0.36$ | $37.5 \pm 0.6$ | $61.7 \pm 5.4$ |
| Reverse LM score ↓ | $94.6 \pm 1.4$ | $80.7 \pm 1.4$ | $106.6 \pm 1.5$ |
| Human evaluation ↑ | $3.49 \pm 0.22$ | $3.27 \pm 0.16$ | $2.78 \pm 0.08$ |
| | LeakGAN-leak | LeakGAN-noleak | LeakGAN-mixed |
| Unique 4grams ↑ | $45k \pm 1.3k$ | $54.4k \pm 2.8k$ | $45.3k \pm 2.4k$ |
| BLEU4 ↑ | $0.219 \pm 0.007$ | $0.193 \pm 0.008$ | $0.21 \pm 0.008$ |
| Self-BLEU4 ↓ | $0.245 \pm 0.01$ | $0.21 \pm 0.009$ | $0.23 \pm 0.011$ |
| FD ↓ | $0.4 \pm 0.009$ | $0.385 \pm 0.02$ | $0.352 \pm 0.008$ |
| LM score ↓ | $67.9 \pm 4$ | $34.9 \pm 1.5$ | $35.9 \pm 1.7$ |
| Reverse LM score ↓ | $114.3 \pm 1.6$ | $87.4 \pm 1.5$ | $99.5 \pm 3.9$ |
| Human evaluation ↑ | $2.47 \pm 0.28$ | $3.35 \pm 0.22$ | $3.22 \pm 0.15$ |

Table 2: Results of best models obtained with our evaluation procedure. For brevity, we report only BLEU4 scores in this table. We have measured scores humans assign to real samples for reference and obtained a value of 4.27. ↓ means lower is better, ↑ higher is better.

Table 2 shows the results obtained by various models using our evaluation procedure. We make the following observations: (i) BLEU scores assign very close values to Conv-LSTM and Language Models, while other metrics massively favor Language Models. Manual inspection reveals that Language Model is a much better model than Conv-LSTM, as shown in Table 1; (ii) vanilla version of SeqGAN performs better that its more advanced versions in our experiments. We attribute this to the fact that we tune our models with respect to Frechet Distance while previous work has optimized BLEU scores. In this work we show that BLEU score is an inadequate metric and it is thus difficult to convincingly say whether one model is better than another based on just BLEU scores.

In addition, we generally observe that hyperparameter search favors low values of generator learning rates. This suggests that lower learning rates help to keep the generator weights close to a Language Model used to initialize the weights. However, we note that BLEU scores of the generated sequences improve suggesting, higher precision for GAN models. We expect metrics that are capable of revealing trade-offs between precision and recall to allow better understanding of what kind of generators GANs learn.

# 7    APPENDIX B: LEAKGAN MODEL

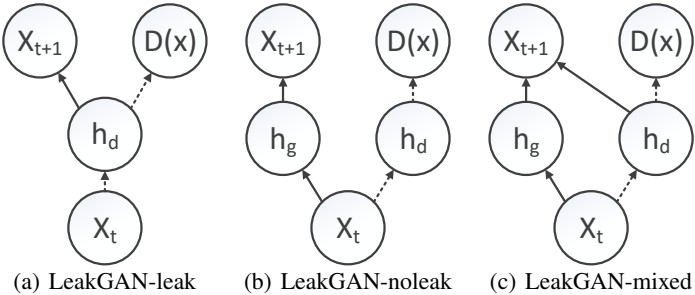

(a) LeakGAN-leak          (b) LeakGAN-noleak          (c) LeakGAN-mixed

Figure 6: Schematic description of the three considered LeakGAN models. Solid and dashed arrows represent weights learned in generator and discriminator phases respectively. $h_g$ and $h_d$ represent hidden states of the generator and discriminator respectively. Note that $h_g$ is absent in LeakGAN-leak case. $x_t$ and $x_{t+1}$ are current and predicted tokens. $D(x)$ is output of the discriminator.

The main motivation behind the LeakGAN models is that the discriminator builds a representation of a sequence in order to tell real sequence from a generated one. It is not obvious that this representation is the same as that of a model trained to predict next word given history would build and thus could be helpful for the generator. A similar observation has been made by Guo et al. (2017). The authors, however, have also introduced a very complicated RL training approach. In addition, they interleave GAN phases where the generator is updated via signal form the discriminator and NLL phases where generator is trained as a conventional Language Model. It is thus not obvious how much does the idea of revealing discriminator's state to the generator actually contribute to the overall result. We thus attempt to decouple this idea from the complicated RL training setup used by the authors and study the utility of passing discriminator's state to the generator during training.

Our initial experiments have shown that it is important to fuse discriminator's and generator's state with a non-linear function and thus we use a one-layer MLP to predict the distribution over next token. In this setup we only use per-step discriminators. Figure 6 shows a graphical representation of one step of three variants of the LeakGAN model.

