# OpenReview forum: "On Accurate Evaluation of GANs for Language Generation"
_ICLR.cc/2019/Conference_

### Official Review · AnonReviewer3 · 2018-11-02
**this paper seems to be timely for this line of work**

**Rating:** 6
**Confidence:** 4

**Review:**

===========================
Since the authors did not provide a proper response to my questions, I have lowered my score from 7 to 6. I think this paper will have a good chance to be a good paper if evaluated more comprehensively, as suggested by reviewers.
===========================

Contributions:

The main contribution of this paper is the study of the currently adopted evaluation metrics for textual GAN models. It was shown that BLEU and Self-BLEU scores used by previous work are insufficient to evaluate textual GAN models, and the authors propose that Frechet Distance and reverse Language Model scores can be a good complement to the above BLEU score evaluations.

Detailed Comments:

(1) Novelty: It seems to me that this paper is timely, as developing GAN models for text generation gains more and more attention in the research community, and it is indeed much needed to provide good evaluation methods. The proposed new metrics seem proper, and the observation that most GAN models do not yield obviously better results than conventional LM is also insightful.

(2) Presentation: This paper is generally well-written and easy to follow. However, when discussing related work in section 3.1, I think one literature [*] is missed. It uses annealed softmax to approximate argmax for textual GAN.

[*] Adversarial Feature Matching for Text Generation, ICML 2017

(3) Evaluation: The experiments are generally well-executed, with some questions listed below.

Questions:

(1) I have some concerns in terms of human evaluation. Though human evaluation is the golden metric, it seems that presenting individual sentences to human raters does not account diversity into consideration. Therefore, systems that generate high quality samples but with less diversity will get a high score in terms of human evalution. Can the authors provide some discussion on this? And if this is the case, how will this change the conclusions in this paper?

(2) I understand why the authors use simplified GAN models for evaluation. However, if the models are not simplified, what the performance will be for LeakGAN and MaskGAN, for example? This seems to be relatively easy to evaluate since the code is open sourced.

Minor issues:

(1) I think the citation format needs to be changed. For example, in many places, it is more natural to use "(Hassan et al., 2018)" than "Hassan et al. (2018)" for example.

---

### Official Review · AnonReviewer2 · 2018-11-03
**Valuable goal, but very limited executions with incorrect claims**

**Rating:** 3
**Confidence:** 4

**Review:**

The paper sets out to improve the evaluation of GAN models as e.g. the previously used BLEU score is not sensitive to semantic deterioration of generated texts. The paper claims to “propose alternative metrics that better
capture the quality and diversity of the generated samples”.


Strengths:
-	The paper has a valuable goal
-	Some of the evaluations are interesting.


Weaknesses:
1.	The claim of the paper to “propose alternative metrics that better capture the quality and diversity of the generated samples” is not met in multiple ways:
a.	The paper seems not to propose any new metrics but evaluate existing ones.
b.	The metrics are not extensively compared to human judgments, e.g. by computing correlation. In fact, Figure 5 suggests that they are not very well correlated.
c.	The diversity is not explicitly studied on generated text samples.
2.	The paper concludes that the human eval “assigns better scores to the Language Model”, which is incorrect as Seq gan scores 3.49 vs. 3.37 for language model (even if the seq gan has higher variance).
3.	The metrics are not very well defined, e.g. with formulas, although this is one of the central points of the paper. e.g. what are the reference the blue score is computed against?

---

> ### Author Response · Authors · 2018-11-14
> **Regarding metrics being not very well defined**
>
> We agree that it would be best to have a self-contained paper, but we have opted for referencing papers introducing metric as they have a much better introduction and discussions of these metrics. The reference used to compute BLEU scores is the full validation set. This is an accepted reference in the language GANs community and one of our goals is to show that it leads to misleading results.

---

> > ### Comment · AnonReviewer2 · 2018-12-11
> > **Author response did not address concerns**
> >
> > The paper was not revised and the major concerns from myself and the limited contribution also pointed out by R1 remain.
> >
> > I thus remain with my original recommendation to reject the paper.

---

### Official Review · AnonReviewer1 · 2018-11-03
**Experimental paper studying an important problem with insufficient / unsurprising conclusions**

**Rating:** 5
**Confidence:** 4

**Review:**

This paper tackles the problem of evaluation of language generation models and particularly focuses on the comparison between GAN-based language models (GAN-LM) vs likelihood-based language models (MLE-LM). Studying the behaviour of current evaluation metrics for language generation as well as finding new ones is an important research topic. I believe that this paper makes a step in the right direction but the magnitude of that step may be insufficient for publication. I appreciate the efforts but I find most of the findings about BLEU not being a good metric and characteristic of reverse PPL rather unsurprising. The majority of the paper is dedicated to describing models / metrics which are well-known instead of performing more solid experimental evaluation (Results start at page 6). Instead, the authors could have focused more on the study of FD for language generation.

-- Details

-"Our main finding is that, when compared carefully, a conventional neural Language Model performs at least as well as any of the tested GAN models", however the authors don't compare with the recent MaskGAN model, which, according to (https://arxiv.org/abs/1801.07736) outperforms MLE variants.

- "We demonstrate that previously used n-gram matching, such as BLEU scores, is an insufficient metric": the fact that BLEU is not ideal for evaluation natural language generation has been pointed out in multiple related papers (e.g. https://arxiv.org/abs/1603.08023) and thus is not surprising.

- "We find that reporting results from the best single run or not performing sufficient tuning introduces significant bias in the reported results": as the authors point out in related works, the variance in GAN results which hinders meaningfulness of the reported results is a also a well-known problem (e.g. https://arxiv.org/pdf/1711.10337.pdf), therefore cannot be considered as a contribution.

- The observed behaviour (sensitivity to mode collapse, word swap, word removal) of the "reverse PPL" metric is pretty much expected, but I agree some experimental results are still interesting.

- On the contrary, I liked the study on the FD metric but I would have loved the paper to focus more on the study of the behaviour of that metric: for example, by examining the robustness under different base models, while the authors only test with the model by Conneau et. al, 2017.

- It would have been good to train a state-of-the-art language model architecture, e.g. AWD-LSTM, and to control regularization. I cannot see if the MLE-LM model is overfitting or not.

-- Style remarks:

- Moving the figures closer to the paragraph where they are described avoids the reader the burden of going back and forth through the paper.

---

> ### Author Response · Authors · 2018-11-14
> **Regarding testing a number of embedding models for use with FD**
>
> We have also experimented with two sequence embedding models from Universal Sentence Encoder (https://arxiv.org/abs/1803.11175)  and found behavior of FD to be generally the same regardless of the base model. The results are presented in Figure 1, row 2. We agree that a better discussion of this point will be helpful.

---

### Author Response · Authors · 2018-11-14
**Author response**

We would like to thank the reviewers for their insightful comments and feedback.

We agree that it is well known that metrics like BLEU have important deficiencies. However, they are still widely used to claim that a given model is SOTA, which is something the community needs to move away from, and we believe further evidence can help. Human evaluations are also not sufficient as it is almost impossible to capture diversity for the unconditional text generation setting, beyond detecting severe mode collapse, as also noted by R3. Our goal was to choose a metric that is capable of capturing both quality and diversity in automated manner.

We agree that it would be nice to have more text GAN models implemented and compared in the same setting. Each model reproduction corresponds to a very large amount of work, and it is not feasible to have one group reproduce all of them. Ideally a community effort would be possible to organize, but that's beyond the scope of this work. We instead focused on some representative publications; the main value of our paper is on pointing out some of the pitfalls of common evaluation protocols and metrics, and propose a methodology that future works can take inspiration from. The  models that we evaluate are examples of the fact that conclusions can be different if one follows a more careful experimental setup.

Regarding comments that some of our contributions have been known to the community, e.g. that reporting single best value is misleading: we agree with that these issues have been raised before, however this approach to evaluation of language GANs is still widely accepted in the literature. In addition, these concerns have been dominantly voiced in the Computer Vision community and has not made their way to the Language Generation works. We are convinced that it is important to explicitly state and demonstrate these issues when working in the Language domain.

---

### Meta-Review · Area_Chair1 · 2018-12-13
**Valuable goal, but execution is somewhat suspect.**

**Confidence:** 3
**Recommendation:** Reject

**Metareview:**

This paper conducts experiments evaluating several different metrics for evaluating GAN-based language generation models. This is a worthy pursuit, and some of the evaluation is interesting.

However as noted by Reviewer 2, there are a number of concerns with the execution of the paper: evaluation of metrics with respect to human judgement is insufficient, the diversity of the text samples is not evaluated, and there are clarity issues.

I feel that with a major re-write and tighter experiments this paper could potentially become something nice, but in its current form it seems below the ICLR quality threshold.